# Automatic and Aligned Anchor Learning Strategy for Multi-View Clustering

## ABSTRACT

Multi-View clustering (MVC) commonly utilizes the anchor technique as a strategic approach to mitigate the computational complexity associated with large-scale data. Existing methods generally assume a pre-selection of anchors to facilitate subsequent clustering tasks. However, the determination of the optimal number of anchors is often non-trivial and necessitates their treatment as a tunable parameter, thereby incurring additional computational overhead. Moreover, it is not reasonable to assume an identical number of anchors across all views, as this assumption restricts the representational capacity of anchors in individual views. To address the above issues, we propose a view adaptive anchor multi-view clustering called Multi-view Clustering with Automatic and Aligned Anchor (3AMVC). We introduce a Hierarchical Bipartite Neighbor Clustering (HBNC) strategy to adaptively select a suitable number of representative anchors from the original samples of each view. Specifically, when the representative difference of anchors lies in a acceptable and satisfactory range, the HBNC process is halted and picks out the final anchors. Moreover, we propose an innovative anchor alignment strategy in response to the varying quantities of anchors across different views. This approach initially evaluates the quality of anchors on each view based on the intra-cluster distance criterion and then proceeds to align based on the view with the highest-quality anchors. The carefully organized experiments well validate the effectiveness and strengthens of 3AMVC.

## CCS CONCEPTS

• **Computing methodologies**; • **Cluster analysis**; • **Theory of computation**; • **Unsupervised learning and clustering**;

## KEYWORDS

Multi-view Clustering, Large-scale Clustering, Anchor Graph Clustering

## 1 INTRODUCTION

Multi-view Clustering (MVC) [13, 39, 40, 50] generally delves into the intricacies of the similarity relationships among sample pairs, and constructs a graph matrix that captures the underlying structure and relationships within the dataset, thereby enabling the accurate completion of the clustering task. MVC typically begins by meticulously constructing individual graph structures for each

distinct view. Subsequently, these distinct graphs are synthesized through a fusion process designed to extract a consensus graph with enhanced quality [8, 20, 36, 42]. For example, Kang *et al.* [8] unified the exploration graph structure, fusion graph and spectral clustering process in one step. Wang *et al.* [36] introduced rank constraints to individual graphs in each view, and then integrated these high-quality graphs to obtain a clearer graph of the consensus clustering structure. Wu *et al.* [42] proposed a voting model for extracting neighbors across views, which can directly obtain clustering labels by skipping the traditional spectral decomposition procedure.

Although a substantial number of research affirming the capability of MVC in enhancing the clustering performance, the sheer scale of data encountered in practical applications still remains a formidable challenge [4, 27]. Traditional MVC methods, which necessitate the construction of graph matrices at a minimum cost of the square of the sample size $n$, are rendered powerless in the face of large-scale datasets [7, 9, 48]. The anchor graph strategy [16, 28, 31, 32, 38, 43, 49, 50] can address the challenges posed by large-scale data application scenarios. Its basic idea is to approximate the overall sample using $m$ anchors on each view. Obviously, the consensus anchor graph has a significant impact on performance. Consequently, a multitude of studies are dedicated to exploring the selection and fusion processes of anchors, aiming to enhance the quality of the consensus anchor graph [32, 37, 44, 50]. K-means becomes the mainstream anchor generation method because of its linear complexity and the ability to preserve the inherent characteristics of the clustering structure. Traditional methods such as [44, 50] use K-means to select anchors on each view and then subsequently fuse them. Sun *et al.* [32] employed the direct optimization of the consensus anchor matrix, replacing the process of individual view optimization followed by the fusion of anchor matrices. Wang *et al.* [37] identified the Anchor-Unaligned Problem (AUP), and performed extensive experiments to demonstrate that the alignment of anchors exerts a profound influence on the performance of clustering.

Despite that the aforementioned approaches have effectively improved the quality of anchor graph and improved the adaptability of multi-view clustering tasks in large-scale data scenarios, they collectively face an limitation, *i.e.*, the selection of anchors necessitates supplementary procedures. The majority of existing approaches employ the correlation number of the cluster count $k$ [14, 35] or the correlation number of the sample size $n$ [12, 28] to judiciously determine the optimal quantity of anchors. Even if the K-means method may preserve the clustering structure to a certain degree, the method of selecting tentative values struggles to approximate the optimal anchors, let alone random sampling. Additionally, although Wang *et al.* has delved into establishing the correct correspondence between anchors across views [37], it does not necessarily select the view with the highest quality anchor

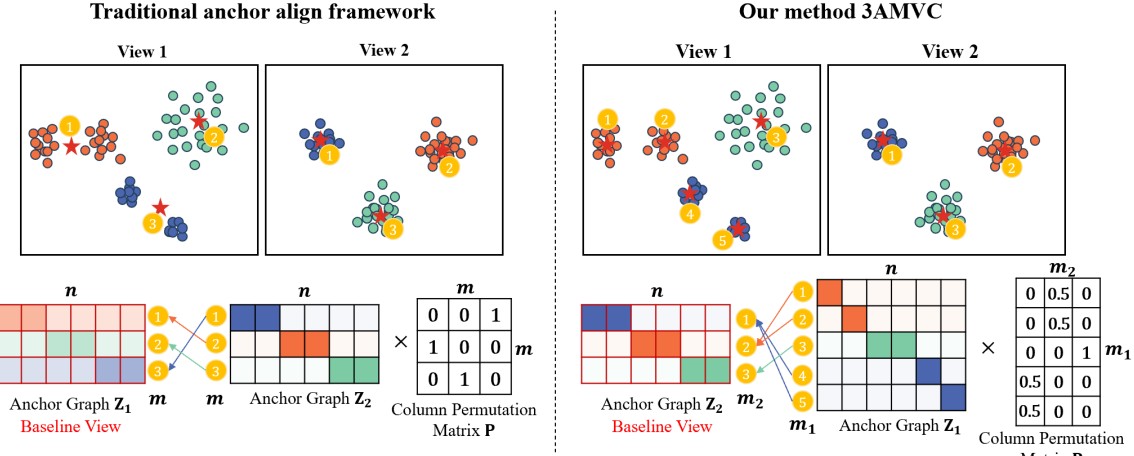

**Figure 1: The framework of the traditional anchor alignment method (Left) and the proposed 3AMVC (Right). The traditional alignment framework fixes the same number of anchors $m$ for each view, and may choose a baseline view with poor quality of anchors. The proposed 3AMVC method adaptively selects anchors of superior quality and accomplishes the alignment task utilizing the baseline view that demonstrates the most exemplary performance.**

graph as the reference for alignment. As far as we know, there exists no universal framework capable of automatically selecting high-quality anchors and aligning them based on the optimal view. This is largely because that it is inherently a challenging problem to perform clustering with an unknown number of clusters on a single view.

In this paper, we propose a general multi-view automatic anchor graph fusion framework termed **A**utomatic and **A**ligned **A**nchor for **M**ulti-view **C**lustering (3AMVC). Specifically, we propose a Hierarchical Bipartite Neighbor Clustering (HBNC) method, which can complete clustering without specifying the number of clusters, so as to automatically obtain high-quality anchors on each view. Then, we improve the alignment method of Wang *et al.* [37], so that it can explore the similarity relationship of anchors between two views when the number of anchors in each view is inconsistent. In addition, we record the similarity relationship between the anchors and the samples during the process of automatically anchor selection, and use it to represent the quality of the anchor graph of the view, so as to determine the optimal view before alignment. Figure. 1 shows the difference between our method and the traditional anchor alignment framework. The traditional alignment framework fails to discern and exploit anchors of superior quality within a single view, and is more likely to perform alignment tasks based on suboptimal views because it lacks a systematic evaluation mechanism for the quality of the anchor graphs. While our method can automatically select better quality anchors on a single view and complete the alignment work based on the view with higher anchor graph quality. We summarize the contributions as follows:

- We propose a novel Hierarchical Bipartite Neighbor Clustering (HBNC) algorithm to identify and select high-quality anchors on a single view. It does not require a prerequisite specification number of anchors and is capable of preserving the inherent characteristics of data structures like K-means.

- We design a general multi-view automatic anchor graph fusion framework termed Multi-view Clustering with Automatic and Aligned Anchor (3AMVC), which can align and fuse anchor graphs according to the view with highest quality anchor graph when the number of anchors is inconsistent.

- Extensive experiments on five benchmark datasets show the effectiveness and efficiency of the proposed method. Compared with the traditional alignment framework FMVACC, the proposed method has improved both in clustering performance and running time.

## 2 RELATED WORK

### 2.1 Multi-View Anchor Graph Clustering

Recently, due to the superior non-linear performance of multi-view graph clustering, it has garnered increasingly extensive research attention [8, 13, 25, 33, 34, 41]. However, the construction of a complete graph matrix requires a computational complexity that is at least quadratic in the number of samples $n$, which often proves inadequate when confronted with large-scale of data scenarios. Therefore, many studies have turned to employ representative anchors for the construction of anchor graphs instead of the complete graphs, which can effectively reduce the quadratic computational complexity to a linear correlation with the number of samples [15, 20, 30, 32, 38, 46].

Specifically, multi-view anchor graph clustering basically follows a two-step strategy of optimization-fusion. The first step is to optimize the anchor graph on each view as follows:

$$\min_{\mathbf{Z}_i} \|\mathbf{X}_i - \mathbf{Z}_i \mathbf{A}_i\|_{\mathrm{F}}^2 + \Omega(\mathbf{Z}_i), \ \text{s.t.} \ \mathbf{Z}_i \geq 0, \mathbf{Z}_i \mathbf{1}_m = \mathbf{1}_n, \quad (1)$$

where $\mathbf{Z}_i \in \mathbb{R}^{n \times m}$ and $\mathbf{A}_i \in \mathbb{R}^{m \times d_i}$ is the anchor graph and the anchor matrix of the $i$-th view respectively. $\mathbf{Z}_i \mathbf{1}_m = \mathbf{1}_n$ is a sum-to-one constraint that ensures that the sum of the sample similarity corresponding to the anchors equals 1.

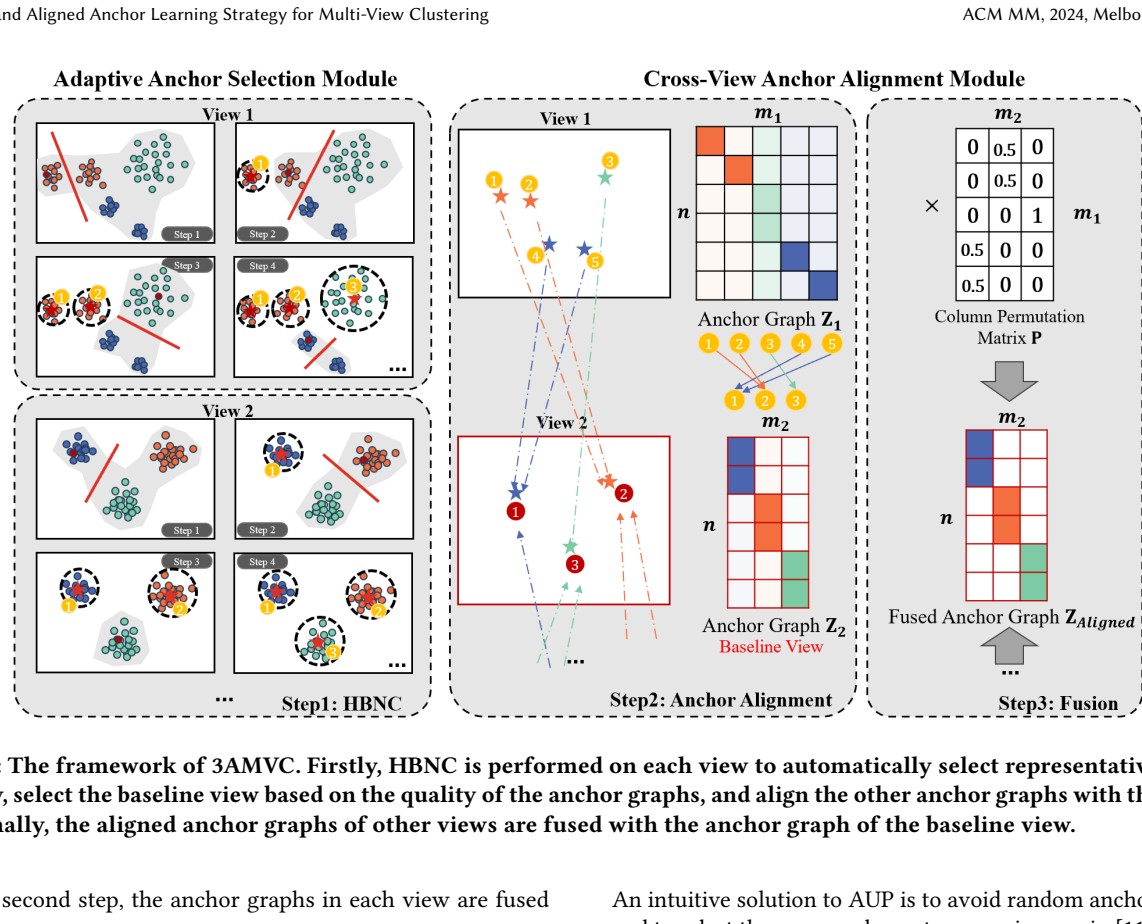

**Figure 2: The framework of 3AMVC. Firstly, HBNC is performed on each view to automatically select representative anchors. Secondly, select the baseline view based on the quality of the anchor graphs, and align the other anchor graphs with the baseline view. Finally, the aligned anchor graphs of other views are fused with the anchor graph of the baseline view.**

In the second step, the anchor graphs in each view are fused through:

$$\min_{\boldsymbol{\beta},\mathbf{S}} \|\beta_p \mathbf{Z}_p - \mathbf{S}\|_F^2, \text{ s.t. } \boldsymbol{\beta} \geq 0, \boldsymbol{\beta}^\top \mathbf{1}_v = 1, \mathbf{S1}_m = \mathbf{1}_n, \quad (2)$$

where $\mathbf{S} \in \mathbb{R}^{n \times m}$ is the fused anchor graph, $\boldsymbol{\beta} \in \mathbb{R}^v$ is the weight coefficient which measures the impact of each view on the fused anchor graph, and $\boldsymbol{\beta}^\top \mathbf{1}_v = 1$ ensures that the sum of weights is 1 and satisfies constraint $\mathbf{S1}_m = \mathbf{1}_n$.

After spectral decomposition of the fusion anchor graph, K-means is performed on the left singular vector to obtain the final clustering result. Multi-view anchor graph clustering can reduce the time and space complexity to $O(vmn)$ and $O(nm^2)$, respectively and effectively adapt to the large-scale data scenarios.

Although the existing approaches can well solve the problem of large-scale multi-view clustering, they all encountered a common problem, that is, the anchors are established before the optimization process commences, hence the quality of these anchors plays a pivotal role to the performance of the clustering. Most existing methods take random sampling and K-means [9, 12] to select anchors, however, both of which require a given number of anchors. This prerequisite number of anchors then becomes a hyperparameter which requires time for searching.

## 2.2 Anchor-Unaligned Problem

The concept of anchors gains traction in the realm of large-scale multi-view clustering. Wang *et al.* [37] have further identified a critical, yet previously overlooked issue within the selection and fusion method which is termed as Anchor-Unaligned Problem (AUP).

An intuitive solution to AUP is to avoid random anchor sampling and to select the same anchor set across views as in [11]. However, this approach restricts the model flexibility and uses the same set of anchors across various views. Both intuitively and existing studies have shown that learning more representative anchors can achieve improved performance over random sampling. Another solution is to optimize the consensus anchors representing cross-view features in a latent common-shared space [32]. Liu *et al.* [16] added constraints to the consensus anchor graph, so that the connectivity of the anchor graph equals to the number of clusters. Nonetheless, employing projections within the common space will inevitably result in the loss of unique, view-specific information. Consequently, the strategic and flexible selection of anchors within a single view, coupled with a fusion process grounded in accurate correspondences, emerges as a superior solution to ensure clustering performance in large-scale scenarian.

Wang *et al.* [37] proposed a Fast Multi-View Anchor-Correspondence Clustering (FMVACC) method to solve AUP. Specifically, FMVACC is to retain the feature and structure information of the anchor set for anchor matching. The optimization formulation can be expressed as:

$$\min_{\mathbf{P}} \|\mathbf{Z}_1 - \mathbf{Z}_i \mathbf{P}\|_F^2 + \lambda \|\mathbf{G}_1 - \mathbf{P}^\top \mathbf{G}_i \mathbf{P}\|_F^2,$$

$$\text{s.t. } \mathbf{P1} = \mathbf{1}, \mathbf{P}^\top \mathbf{1} = \mathbf{1}, \mathbf{P} \in \{0, 1\}^{m \times m}, \quad (3)$$

where $\|\mathbf{Z}_1 - \mathbf{Z}_i \mathbf{P}\|_F^2$ is the feature correspondence and $\|\mathbf{G}_1 - \mathbf{PG}_i\mathbf{P}\|_F^2$ is the structure correspondence. Moreover, $\lambda$ is a parameter for balancing two terms, $\mathbf{G}$ denotes the graph structure inside the anchor set, $\mathbf{Z}$ is the respective anchor graph which represents the

similarities between the samples and the anchors in each view, $i \in \{2, 3, ..., v\}$ represents the other views expect the 1-st one.

The above formulation indicates that the anchor graphs in the other views are aligned to the anchor graph in the 1-st view from the feature and structure correspondence simultaneously. While FMVACC addresses the cross-view alignment of anchors with different dimensions, it still falls short in thoroughly exploring view with superior quality, hence it simplistically adopts the first view as the baseline for alignment tasks. Should the anchor graph derived from the 1-st view fail to capture an precise representation of the underlying structure, and if the alignment between the anchor and the sample is not close enough, then there exists a risk. Specifically, it could lead to subsequent views aligning based on a flawed relationship. Furthermore, since the existing anchor graph clustering methods all use the same number of anchors across views, FMVACC does not consider the scene where the number of anchors is different across views, so it lacks the capability to effectively align disparate anchors.

## 2.3 Clustering without Knowing Cluster Number

At present, the prevailing array of methods inherently assume the number of clusters as a pre-defined parameter within the clustering process. For instance, the K-means and k-NN methods, which are widely applied throughout clustering procedures [5, 17–19, 26, 47], are both derived from the premise of a predefined number of clusters $k$. Moreover, the task of selecting representative anchors on a single view is essentially clustering without knowing the number of clusters. An effective solution is to use other evaluation criteria instead of cluster number $k$. Hierarchical clustering [22, 23] and density-based clustering [1, 10] are representative methods of such solution. Hierarchical clustering gradually expands the clustering granularity based on distance measure, starting from the recent two samples clustered into one class, until all samples clustered into one class, and the stopping condition is an appropriate number of clusters or a distance threshold. And density-based clustering generally complete clustering based on the density threshold. There are other methods that design models and propose new criteria to regulate samples within clusters [2, 6]. In summary, the aforementioned approaches are essentially provide criteria for the number of clusters, either in a directly or indirectly manner. Nevertheless, ascertaining the optimal number of clusters based solely on the intrinsic characteristics of the dataset, without presupposing any artificial thresholds, presents a significantly more intricate challenge. Ronen *et al.* [29] incorporated a subclustering network based on the deep clustering frameowrk to ascertain an optimal number of clusters $k$. While Menon *et al.* [21] proposed to reflect the appropriate number of clusters by the drastic change of clustering scores. Aiming at the anchor selection, we prefer the clustering method without knowing either cluster number or other alternative indicators, whose computational complexity is affordable.

## 3 THE PROPOSED METHOD

In this section, we introduce the method for selecting high-quality anchors on a single view, and elaborate the alignment strategies of these anchors in relation to the baseline view. Figure. 2 shows the

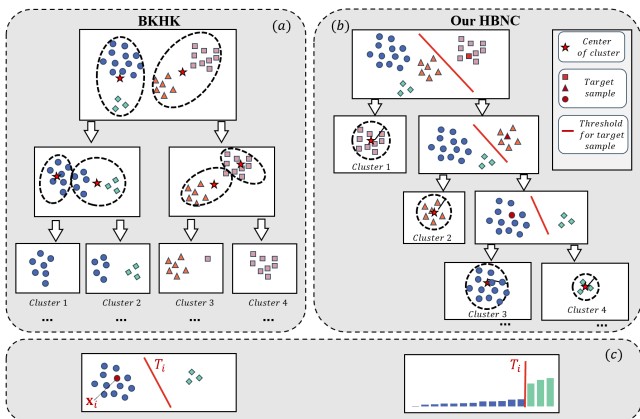

**Figure 3: The anchor selection strategy of BKHK and our HBNC. (a) BKHK performs binary classification of samples which leads to suboptimal performance when the number of samples across different classes is imbalanced. (b) Our method can flexibly select neighbors based on the distances between samples, effectively addressing scenarios where the distribution of sample sizes across clusters is uneven. (c) Specifically, we evaluate the distance between the target sample and other samples within the sample range to be clustered, and find the optimal threshold.**

overall framework of our Multi-view Clustering with Automatic and Aligned Anchor (3AMVC). Our method takes three steps: i) an Adaptive Anchor Selection Module to select the appropriate anchors on a single view, ii) a Cross-View Anchor Alignment Module to align the views based on the view with best quality of the anchor graph, and iii) finally performing fusion to the aligned views.

## 3.1 Adaptive Anchor Selection Module

The fundamental premise of our anchor concept is to effectively address the challenges posed by large-scale data so as to make the computational complexity involved in the selection of anchors within a single view affordable. Balance K-means based Hierarchical K-means (BKHK) method [52] offers a novel approach by combining the flexibility of hierarchical clustering with a relatively low computational complexity, providing us with a fresh perspective on tackling these challenges. However, BKHK imposes excessively stringent demands on the equalization performance of the dataset, and it specifies a termination condition based on the number of clusters in the hierarchical clustering process.

Building upon the foundation of BKHK, we introduce an effective Hierarchical Bipartite Neighbor Clustering (HBNC) method that enable it to better accommodate datasets with uneven intra-cluster sample sizes and to automatically determine the optimal termination point. Figure. 3 shows the specific process of BKHK and our HBNC. BKHK classifies all samples into two categories layer by layer based on the distance between samples, and finally obtains $k$ clusters with balanced number of internal samples. However, real data sets often do not follow the rules of intra-cluster equilibrium, and BKHK will fail in such cases such as part (a) of Figure. 3. In comparison, our method can determine the relevant

optimal threshold, so as to flexibly divide the samples inside and outside the target cluster.

### 3.1.1 How to determine the optimal threshold?

Intuitively, the relationships among samples within the same cluster should be markedly distinct from those between samples belonging to different clusters. As depicted in Figure 3 (c), there is a significant disparity in the distances between the target sample and the samples outside the cluster compared to those within its own cluster.

Based on this principle, we design a new criterion for optimizing the optimal threshold to determine the neighbors of the target sample in the range of sample to be clustered. Take the $l$-th layer clustering process as an example, we first randomly select a target sample $\mathbf{x}_i$ from $n_l$ samples to be clustered, and calculate the distance between the target sample and other samples denoted as $\mathbf{d} \in \mathbb{R}^{n_l \times 1}$. After sorting $\mathbf{d}$, we get the vector form as $\hat{\mathbf{d}} = \left[ d_{i[1]}, d_{i[2]}, ..., d_{i[n_l]} \right]^\top$. Our goal is to find an optimal threshold $T_i$ related to the sample $\mathbf{x}_i$, so as to divide the nearest neighbors of the target sample. The optimal threshold $T_i$ should satisfy the following condition:

$$d_{i[1]} \leq d_{i[2]} \leq ... d_{i[k_i]} \leq T_i \leq d_{i[k_i+1]} ... \leq d_{i[n_l]}. \quad (4)$$

In this paper, we design an ingenious formula to determine the appropriate threshold $T_i$. Inspired by the grayscale binarization in the field of image segmentation [24], we first introduce the concept of inter-class error. Regardless of the layer in which it is performed, the clustering task at hand involves categorizing the samples into two groups. For convenience, we consider the cluster containing the target sample as the first cluster, and all other samples as the second cluster. We represent the proportion of each type of sample within the entire samples as $p_1$ and $p_2$, as well as the average distance between each cluster of samples and the target sample as $\mu_1$ and $\mu_2$. Then the inter-class error can be expressed as:

$$p_1(\mu_1 - \mu)^2 + p_2(\mu_2 - \mu)^2, \quad (5)$$

where $\mu$ is the mean value of all samples.

The proportion and mean satisfy the following conditions: $p_1 + p_2 = 1$ and $p_1\mu_1 + p_2\mu_2 = \mu$. Based on that, we can transform Eq. (5) as follows:

$$p_1 p_2 (\mu_1 - \mu_2)^2. \quad (6)$$

Our expectation for the optimal threshold is that a threshold can make the distance between samples in the cluster smaller and at the same time the inter-class error larger. Therefore, the criterion for determining the optimal threshold can be formulated as follows:

$$\min_{k_i} \frac{\mu_1}{p_1 p_2 (\mu_1 - \mu_2)^2}, \quad (7)$$

where $k_i$ is the number of samples in the first cluster with the target sample.

Since the number of neighbors of the target sample is discrete and finite, constrained within $\{1, 2, ..., n_l\}$, the value of Eq. (7) is also discrete and finite. Consequently, by determining the $k_i$ neighbors of the target sample $\mathbf{x}_i$ that can minimize the Eq. (7), we can obtain the optimal threshold $T_i \in \left[ d_{i[k_i]}, d_{i[k_i+1]} \right)$. In practice, we only need $T_i$ as an exact value, so for the sake of simplicity, we let

---

**Algorithm 1** Bipartite neighbor clustering of the $l$-th layer

**Input:** Samples to be clustered $\mathbf{X}_l \in \mathbb{R}^{d \times n_l}$.
1: Randomly select a sample as the target sample $\mathbf{x}_i$.
2: Calculate the distance $\mathbf{d}_l$ between all samples and the target sample.
3: Sort the vector $\mathbf{d}_l$ to get $\hat{\mathbf{d}}_l$.
4: Calculate the neighbors and the optimal threshold of the target sample according to Eq. (7).
**Output:** The neighbors and the optimal threshold of the target sample $\mathbf{x}_i$.

---

$T_i = d_{i[k_i]}$. We summarize this bipartite neighbor clustering process in the $l$-th layer as Algorithm 1.

### 3.1.2 How to determine the number of clusters?

It is widely recognized that the closer the samples within a cluster are to the clustering center, the more concentrated the distribution of samples in that cluster, and consequently, the more representative the cluster center becomes. K-means clusters according to this principle. For a specific cluster $i$, the problem can be expressed as:

$$\sum_{\mathbf{x}_j \in \Theta_i} \|\mathbf{x}_j - \theta_i\|_2^2, \quad (8)$$

where $\Theta_i$ denotes the $i$-th cluster, $\mathbf{x}_j \in \Theta_i$ represents its samples, $\theta_i$ denotes the center of the cluster and $\theta_i = \frac{1}{c_i} \sum_{j=1}^{c_i} \mathbf{x}_j$ with $c_i$ being the number of samples in this cluster.

The proposed Hierarchical Bipartite Neighbor Clustering (HBNC) algorithm introduces a new cluster through the binary classification task of each layer. Consequently, if our algorithm performs through $L$ layers, then similar to K-means, the intra-cluster distance of all clusters in our algorithm is expressed as:

$$\sum_{i=1}^{L} \sum_{\mathbf{x}_j \in \Theta_i} \|\mathbf{x}_j - \theta_i\|_2^2. \quad (9)$$

Although our HBNC algorithm operates without a predetermined number of clusters, by effectively managing the stopping criteria of the algorithm, we can halt the clustering process at an opportune moment, thereby ascertaining the appropriate number of clusters. The clustering result should lead to the centers of all clusters being highly representative. Therefore, at the beginning of the clustering task at each layer, we select the range of the sample to be clustered based on the criterion that the distance between the center and the sample within the cluster is maximized for the chosen cluster. It is imperative to avoid obtaining many clustering centers to represent outliers, and avoid huge differences in the value of Eq. (8) of each cluster. Here, at most one cluster center representing outliers is acceptable. Therefore, when a cluster is with a sample size of 1 again or the target value of the Eq. (8) of all clusters tends to average, we terminate the algorithm and obtain a more representative anchor set. The complete HBNC process ise elaborated in Algorithm 2.

As for the complexity of the algorithm, the computational complexity of each layer of HBNC is to calculate the distance of all samples within the sample range of the layer, which equals to $O(n_l d)$. Consequently, the complexity of the whole HBNC is $O((n_1 + n_2 + ... n_L)d)$, where $L$ is the number of layers. Relative to the first layer, which processes the entire dataset of $n_1 = n$ entries, the computational complexity of the subsequent layers is significantly reduced.

---

**Algorithm 2** Anchor selection stratrgy based on HBNC.

---

**Input:** Dataset $\mathbf{X} \in \mathbb{R}^{d \times n}$.
1: **while** Not converge **do**
2:    Update the layer $l$.
3:    Determine the range of samples to be clustered $\mathbf{X}_l \in \mathbb{R}^{d \times n_l}$.
4:    Complete the bipartite neighbor clustering of the $l$-th layer according to Algorithm 1.
5:    Calculate the intra-distance of each cluster according to Eq. (8).
6: **end while**
**Output:** Anchor quality according to Eq. (9) and the anchors $\Theta$.

---

As such, the overall complexity of our HBNC algorithm can be denoted as $O(nd)$.

## 3.2 Cross-View Anchor Alignment Module

The unequal number of anchors automatically selected on each view compels us to further explore one-to-many or many-to-one matching relationships, rather than being constrained by the strict one-to-one matching framework. In addition, matching based on the view where the higher quality anchor set is located should bring better clustering performance.

### 3.2.1 How to determine the baseline view for alignment?

In the previous section, we propose a formulation in Eq. (9) to simultaneously adeptly capture the relationship between the global clustering center and the samples and reflect the representative performance of the clustering center essentially. In this section, we therefore employ Eq. (9) as a metric of anchor representativeness to assess the quality of the anchor graph for each individual view. In the process of devising anchor quality metrics for views, we discern that the summation approach offers numerous advantages over the mean-based formulations. Since our ultimate goal is to perform spectral clustering on the fusion anchor graph, when the distribution of a certain type of samples forms a ring shape, it is preferable that the selected anchors also follow a ring pattern, rather than being at the center of the ring. For the sample-center distance in the same cluster like Eq. (8), sum rather than average, can avoid the ring distribution of the samples associated with the selected anchor. In addition, it is clear that under the same sample distribution, the more the number of anchors, the stronger the average representation of each anchor. Considering the variability in the number of anchors selected for each view, relying on an average-based calculation may inadvertently place views with fewer anchors at a relative disadvantage. By applying a summation technique universally, the representativeness of the anchors is effectively distributed among all samples, making it a more advantageous strategy. Ultimately, we select the view with the smallest value from Eq. (9), where the anchors exhibit the strongest representativeness and all samples are closer to their respective anchors, to serve as the baseline view.

### 3.2.2 How to align anchor set with unequal number?

According to Wang *et al.*[37], we can align the anchors in terms of feature and structure by summing the values of the anchors in different dimensions. In order to match the optimal view and our inconsistent number of anchors, our improved alignment method

**Table 1: Multi-view Datasets in our Experiments**

| Dataset | Size | Clusters | Views |
|---|---|---|---|
| ForestTypes | 523 | 4 | 3 |
| Reuters | 1200 | 6 | 5 |
| MFeat | 2000 | 10 | 2 |
| Caltech256 | 30607 | 256 | 4 |
| VGGFace2 | 36287 | 100 | 4 |

is expressed as follows:

$$\min_{\mathbf{P}} \|\mathbf{Z}_b - \mathbf{Z}_i \mathbf{P}\|_{\mathbf{F}}^2 + \lambda \|\mathbf{G}_b - \mathbf{P}^\top \mathbf{G}_i \mathbf{P}\|_{\mathbf{F}}^2,$$
$$\text{s.t. } \mathbf{P1} = \mathbf{1}, \mathbf{P}^\top \mathbf{1} = \mathbf{1}, \mathbf{P} \in \{0,1\}^{m_i \times m_b}, \quad (10)$$

where $\mathbf{Z}_b \in \mathbb{R}^{n \times m_b}$ is the anchor graph of the baseline view, $\mathbf{G}_b$ and $\mathbf{G}_i$ represent the inner graph structures of the baseline view and the $i$-th view. It is noteworthy that the matching matrix $\mathbf{P}$ here is specific for each view and can express a one-to-many or many-to-one relationship about the baseline view.

After obtaining $\{\mathbf{P}\}_{i=1,i\neq b}^v$, we can obtain the fused aligned anchor graph based on the baseline view as:

$$\mathbf{Z}_{Aligned} = (\mathbf{Z}_b + \sum_{i=1,i\neq b}^v \mathbf{Z}_i \mathbf{P}_i)/v. \quad (11)$$

Ultimately, by performing spectral decomposition on $\mathbf{Z}_{Aligned}$ according to Eq. (11), and subsequently applying K-means clustering to its left singular values, the final clustering results are successfully derived.

## 3.3 Complexity Analysis

In terms of space complexity, our major memory costs are matrices $\{\Theta_i\}_{i=1}^v \in \mathbb{R}^{m_i \times d_i}$, $\{\mathbf{P}_i\}_{i=1,i\neq b}^v \in \mathbb{R}^{m_i \times m_b}$, $\{\mathbf{Z}_i\}_{i=1}^v \in \mathbb{R}^{n \times m_i}$. The space complexity of 3AMVC is $O(m_{max}d + \sum_{i=1}^v m_i n + m_b^2 v)$. $m_{max}$ is the largest number of anchors in all views. $m_b$ is the number of anchors in the baseline view. In large-scale scenarios, the main amount of computation is mainly the correlation term of $n$. Therefore, the space complexity of 3AMVC is $O(n)$.

As for time complexity, which consists of three steps. As mentioned above, the time complexity of HBNC on a single view is $O((n_1 + n_2 + ... n_L)d)$. In the alignment phase, updating $\{\mathbf{Z}_i\}_{i=1}^v$ costs $O(nm_i d_i)$. Updating $\{\Theta_i\}_{i=1}^v$ needs $O(nm_i m_b + m_i m_b d)$. Then the alignment module costs $O(m_b^2 m_i)$. After obtaining the aligned and fused anchor graph, SVD needs $O(nm_b^2)$. Consequently, the time complexity of our 3AMVC is approximately $O(n)$, which can better cope with large-scale data scenarios.

## 4 EXPERIMENT

## 4.1 Experiment Setting

We verify the effectiveness of our algorithm on five widely used multi-view benchmark datasets: ForestTypes, Reuters, MFeat, Caltech256, and VGGFace2. The information of these datasets is listed in Tab. 1. We compare our method with 8 state-of-the-art multi-view clustering methods: BMVC [51], LMVSC [9], FMCNOF [45], FPMVS-CAG [38], EOMSC-CA [16], FMVACC [37], UDBGL [3], and fastMICE [4].

**Table 2: Empirical comparison of our 3AMVC with nine baseline methods on five benchmark datasets.**

| Methods | BMVC | LMVSC | FMCNOF | FPMVS-CAG | EOMSC-CA | FMVACC | UDBGL | fastMICE | Proposed |
|---------|------|-------|--------|-----------|----------|--------|-------|----------|----------|
| | | | | | ACC | | | | |
| ForestTypes | 0.7266 | 0.7921 | 0.4551 | 0.7222 | 0.7419 | 0.7929 | 0.4990 | 0.7166 | 0.7984 |
| Reuters | 0.4950 | 0.4792 | 0.2858 | 0.4582 | 0.4758 | 0.5532 | 0.2783 | 0.2856 | 0.5734 |
| MFeat | 0.6905 | 0.5690 | 0.5695 | 0.6511 | 0.8220 | 0.8322 | 0.1575 | 0.8285 | 0.8737 |
| Catech256 | 0.0871 | 0.0957 | 0.0270 | 0.0891 | 0.0985 | 0.0911 | 0.0776 | 0.0942 | 0.1023 |
| VGGFace2 | 0.0617 | 0.0623 | 0.0347 | 0.0634 | 0.0858 | 0.0668 | 0.0511 | 0.0508 | 0.0756 |
| | | | | | NMI | | | | |
| ForestTypes | 0.5338 | 0.5446 | 0.1699 | 0.4820 | 0.4724 | 0.5533 | 0.3116 | 0.5171 | 0.5397 |
| Reuters | 0.2946 | 0.2786 | 0.0715 | 0.2459 | 0.2524 | 0.3377 | 0.1225 | 0.1913 | 0.3316 |
| MFeat | 0.6653 | 0.8246 | 0.5547 | 0.5777 | 0.8319 | 0.7241 | 0.1990 | 0.8362 | 0.8240 |
| Catech256 | 0.3158 | 0.3196 | 0.0000 | 0.2343 | 0.2468 | 0.2990 | 0.2375 | 0.3235 | 0.3210 |
| VGGFace2 | 0.1426 | 0.1191 | 0.0581 | 0.1282 | 0.1578 | 0.1300 | 0.1013 | 0.1040 | 0.1404 |
| | | | | | Fscore | | | | |
| ForestTypes | 0.6729 | 0.6572 | 0.4638 | 0.5828 | 0.6057 | 0.6460 | 0.5127 | 0.6014 | 0.6752 |
| Reuters | 0.3633 | 0.3570 | 0.2411 | 0.3487 | 0.3367 | 0.4013 | 0.3002 | 0.3320 | 0.4061 |
| MFeat | 0.6137 | 0.7811 | 0.4552 | 0.5250 | 0.7701 | 0.7078 | 0.2046 | 0.7837 | 0.7986 |
| Catech256 | 0.0663 | 0.0600 | 0.0117 | 0.0322 | 0.0336 | 0.0664 | 0.0148 | 0.0738 | 0.0792 |
| VGGFace2 | 0.0277 | 0.0251 | 0.0239 | 0.0314 | 0.0322 | 0.0267 | 0.0217 | 0.0201 | 0.0297 |

We set their parameters within the recommended range for all comparison algorithms aforementioned. In the proposed method, we adjusted $\beta$ to $[10^{-2}, 1, 10^2]$, $\lambda$ to $[0, 10^{-4}, 10^{-2}, 1, 10^4]$. To assess the clustering performance, we employ three well-used criteria consisting of accuracy (ACC), normalized mutual information (NMI) and Fscore.

### 4.2 Experimental Results

Table. 2 reports the clustering results of other methods and our 3AMVC on five benchmark datasets. The best results are marked in red, while the second-best results are marked in blue. According to the results, we have the following observations:

- Our method exceeds other algorithms in most indicators, or reaches a considerable level with other algorithms. In terms of the ACC metric, 3AMVC achieves improvements of 0.69% (ForestTypes), 3.65% (Reuters),4.97%(MFeat) 12.29%(Catech256), and 13.17%(VGGFace2) respectively, when compared to the troditional alignment algorithm FMVACC.
- On other datasets, we also obtain comparable results with the suboptimal algorithm, and it is worth noting that almost all the algorithms compared require the pre-K-means process to select the anchors, while our algorithm avoids the process of searching for the optimal number of anchors.

### 4.3 Parameter Sensitivity Analysis

Our experimental setup involves two hyperparameters, namely, $\beta$ and $\lambda$. As shown in Figure. 4, we use the ACC metric to show the influence of two parameters on the clustering effect on two datasets: ForestTypes and Caltech256.

We observe that the influence of parameter changes on the clustering effect is not intensive. Relatively speaking, when $\beta$ is set as

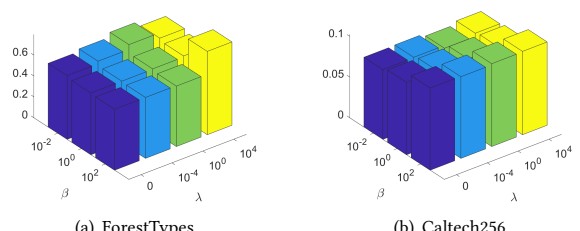

(a) ForestTypes       (b) Caltech256

**Figure 4: Sensitivity Analysis on five dataset with ACC metric.**

$10^2$ and $\lambda$ is set as $10^4$, the clustering performance will be further improved.

### 4.4 Visualization

In order to illustrate the ability of our 3AMVC in anchor selection, we employ visual representations to display the selected anchors in the single view. We run K-means and our HBNC algorithm on the two views of MFeat to select the appropriate anchors, and the results are shown in Figure. 5. The samples are represented by points, and the selected anchor points are uniformly represented by red stars.

From Figure. 5, we observe that although K-means can better cluster sample points into $k$ clusters based on characteristics, the approach, which utilizes cluster centers as representatives for anchor points, struggles to preserve the inherent graph structure within the clusters. Our HBNC algorithm effectively secures a set of more representative anchor points without the need to predefine the number of clusters. On one hand, the samples represented by these

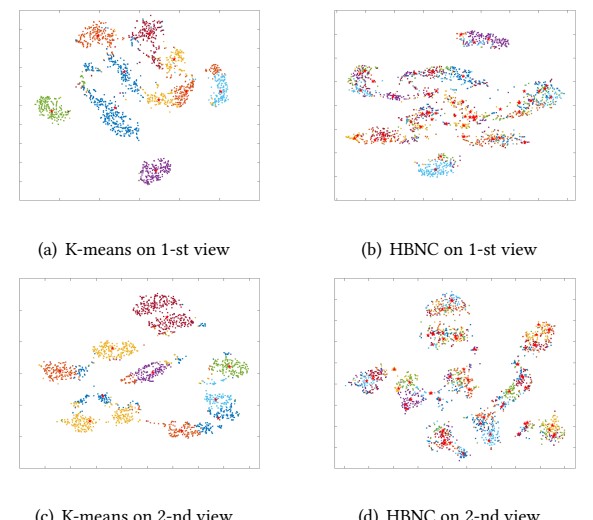

(a) K-means on 1-st view      (b) HBNC on 1-st view

(c) K-means on 2-nd view      (d) HBNC on 2-nd view

**Figure 5: Visualization of selected anchor points on MFeat dataset.**

**Table 3: Experiment results of ablation studies**

| Datasets | Methods | ACC | NMI | Fscore |
|---|---|---|---|---|
| ForestTypes | $3A\_neither$ | 0.7231 | 0.4227 | 0.5628 |
|  | $3A\_w/o\_SA$ | 0.6161 | 0.4502 | 0.5540 |
|  | $3A\_w/o\_AB$ | 0.7217 | 0.4187 | 0.5604 |
|  | $3AMVC$ | **0.7984** | **0.5397** | **0.6752** |
| Reuters | $3A\_neither$ | 0.4367 | 0.2312 | 0.3153 |
|  | $3A\_w/o\_SA$ | 0.4199 | 0.2070 | 0.2970 |
|  | $3A\_w/o\_AB$ | 0.5509 | 0.3282 | 0.3886 |
|  | $3AMVC$ | **0.5734** | **0.3316** | **0.4061** |
| MFeat | $3A\_neither$ | 0.7676 | 0.6993 | 0.6618 |
|  | $3A\_w/o\_SA$ | 0.7744 | 0.7032 | 0.6664 |
|  | $3A\_w/o\_AB$ | 0.8176 | 0.7874 | 0.7312 |
|  | $3AMVC$ | **0.8736** | **0.8240** | **0.7986** |
| Caltech256 | $3A\_neither$ | 0.0882 | 0.2951 | 0.0591 |
|  | $3A\_w/o\_SA$ | 0.0922 | 0.2980 | 0.0692 |
|  | $3A\_w/o\_AB$ | 0.0485 | 0.2516 | 0.0203 |
|  | $3AMVC$ | **0.1023** | **0.3210** | **0.0792** |
| VGGFace2 | $3A\_neither$ | 0.0651 | 0.1267 | 0.0256 |
|  | $3A\_w/o\_SA$ | 0.0713 | 0.1371 | 0.0282 |
|  | $3A\_w/o\_AB$ | 0.0633 | 0.1262 | 0.0244 |
|  | $3AMVC$ | **0.0756** | **0.1404** | **0.0297** |

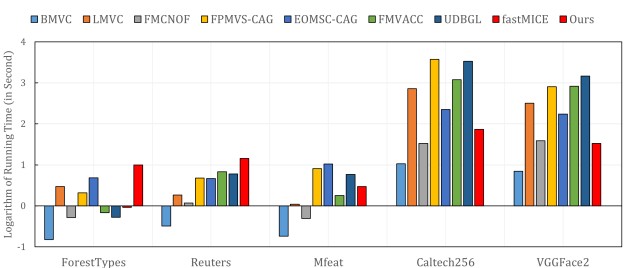

**Figure 6: Running time for all algorithms.**

anchors exhibit a higher degree of similarity to them, which in turn more accurately captures the graph's structure. On the other hand, this approach also circumvents potential inaccuracies that may arise from anchors being positioned in the interstitial gaps between samples. On the MFeat dataset, HBNC finally selected 54 and 64 anchors on the two views respectively.

### 4.5 Ablation Studies

In order to illustrate the effectiveness of our automatic anchor selection strategy and baseline view alignment strategy, we conducted ablation experiments on these two parts respectively. The experimental results are shown in Table 3. Methods that do not use either strategy are recorded as $3A\_neither$. $3A\_w/o\_SA$ means that the anchors are not automatically selected, and we use the K-means method instead. $3A\_w/o\_AB$ stands for not aligning based on the baseline view, we align based on the first view uniformly. $3AMVC$ means the complete algorithm. We conduct experiments under the best parameter settings of each dataset. From the Table 3, we find that the performance of $3A\_w/o\_SA$ is significantly improved on large datasets. This fully demonstrates the the importance of high-quality anchor selection for improving large-scale clustering performance. In addition, we found that the performance of $3A\_w/o\_SA$ is lower than $3A\_neither$ on some datasets, which we believe is due to the mismatch between the anchor points selected by K-means and our view quality metrics.

### 4.6 Complexity

Our method can be well applied to large-scale scenes. We record our time comparison with other algorithms and draw Figure 6. Compared with FMVACC, our algorithm has lower running time. We speculate that this may be due to the computational complexity of the K-means anchor selection strategy in each iteration is $n$, and

the computational complexity of our algorithm will decrease when iterating again.

## 5 CONCLUSION

In this paper, we introduce a novel Multi-view Clustering with Automatic and Aligned Anchor (3AMVC) method designed to address challenges of multi-view clustering in large-scale data scenarios. The algorithm initiates with the innovative Hierarchical Bipartite Neighbor Clustering (HBNC) approach, which eliminates the need for a predefined number of clusters. This method identifies and selects representative anchors within a single view, enabling the construction of a robust anchor graph without relying on prior knowledge. The subsequent phase involves the alignment of all anchor graphs to the baseline view exhibiting the highest anchor graph quality, culminating in the formation of a consolidated, high-quality fused anchor graph. Our proposed method stands out in the current research landscape by offering an effective parameter-free strategy for anchor point selection. Extensive experiments verify the effectiveness and efficiency of our proposed 3AMVC.

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
