# OpenReview forum: "Automatic and Aligned Anchor Learning Strategy for Multi-View Clustering"
_acmmm.org/ACMMM/2024/Conference — MM2024 Poster_

### Official Review · Reviewer_XUZj · 2024-05-12

**Rating:** 3
**Confidence:** 3

**Summary:**

Traditional multi-view clustering methods often suffer from scalability due to the quadratic time complexity. The anchor graph is a great strategy to address this challenge in which the quality of anchors is crucial for the performance. This paper proposes a novel method to adaptively select high-quality anchors for each view to achieve better clustering performance. The author further solves the anchor unaligned problem especially when the number of anchors in different views may not agree. Experiments on several datasets demonstrate certain superiority of 3AMVC compared to existing methods.

**Strengths:**

3AMVC may be the first work to select different numbers of anchors within each view by introducing and modifying hierarchical clustering. Inter- and intra-cluster distances are employed to determine the partition border, which is impressive. The authors further modify the formula proposed by Wang et al. to handle the unaligned anchors of various numbers. Extensive ablation studies prove the effectiveness of the proposed modules.

**Limitations:**

1.	It would be better to explain the optimization process of Eq. (10) more explicitly.
2.	The comparison with nine baseline methods on five benchmarks may not be convincing enough to demonstrate the superiority of 3AMVC. Hypothesis testing would be needed if possible.
3.	I wonder why 3AMVC takes more time on smaller datasets like ForestTypes and Reuters.

**Suitability:**

3

---

### Official Review · Reviewer_C1kY · 2024-05-23

**Rating:** 3
**Confidence:** 3

**Summary:**

To avoid the additional parameter search steps caused by determining the optimal number of anchors in large-scale multi-view clustering tasks, this paper proposes a multi-view clustering method named 3AMVC that can automatically select automatic and aligned anchors. Specifically, the improved HBNC method is used to select the representative anchors on each view, and then the aligned consensus anchor graph is obtained by the fusion of all view anchor graphs according to the most representative one.

**Strengths:**

1- This paper boasts a well-organized structure and coherent logic.
2- The effectiveness of the proposed method is meticulously verified through an extensive array of experiments.

**Limitations:**

1- In Figure 1, why cannot this method choose view 1 as the baseline view? If view 1 is selected as the baseline view, will the clustering performance be affected due to the inconsistency between the number of anchors and the number of clusters?
2- From Figures 1 and 2, it is observed that the elements in the permutation matrix P are decimals ranging from 0 to 1. However, in the final target equation (10), the matrix P's values are integers, specifically 0 or 1, please explain this phenomenon.
3- This paper proposes a novel HBNC method to select the most representative anchors. Specifically, the OTSU method is applied to the distance between samples, so as to find the optimal threshold for the target sample to complete the binary classification. However, the distance metric is greatly affected by the target sample. Changes in the target sample will lead to changes in the overall distance vector. This paper adopts a random method to select the target sample. This raises the question of whether the experimental results are significantly affected by instability.
4- It is widely acknowledged that the key to the algorithm with the unknown number of clusters is to determine the stopping condition, because the stopping time is closely linked to the final number of clusters. While Algorithm 2 provides a comprehensive depiction of HBNC's process for the specific selection of anchors, it falls short of delineating the pivotal stopping conditions. And there is no convergence proof of HBNC in this paper. Please further explain how to determine the number of clusters on a single view.
If the authors can address my concerns above, I would be happy to increase the rating.

**Suitability:**

3

---

### Official Review · Reviewer_YTja · 2024-05-23

**Rating:** 4
**Confidence:** 4

**Summary:**

This paper proposes a novel multi-view clustering method named 3AMVC, which not only skillfully combines the advantages of automatic anchor selection and baseline view alignment, but also finally obtains a high-quality consensus anchor graph for spectral clustering. There are two main innovations in this paper. First, different from traditional anchor selection methods, the framework this paper proposed 3AMVC does not require specifying the number of anchors. Second, it improves the traditional anchor alignment framework so that it can complete the fusion according to the view with the best quality of the anchor graph.

**Strengths:**

(1) In this paper, the author has created a delicate and clear illustration that effectively presents the central concepts introduced.
(2) The paper thoroughly elucidates the rationale and efficacy of the proposed method.
(3) The writing is notably smooth and clear.

**Limitations:**

(1) In this paper, the anchor selection process and threshold determination process of HBNC are given in detail, but HBNC is only a module of 3AMVC, and the whole algorithm is not comprehensively summarized in this paper. It should be considered to add the algorithm.
(2) The formula 9 designed by the author is essentially used to measure the representative performance of anchors. The representativeness of the anchor does not directly correlate with the quality of the anchor. However, is it reasonable to use the representative performance of anchors as an evaluation criterion for the quality of anchor graphs in the single view?
(3) From the experimental results of running time, 3AMVC does not achieve the best results. It even has the longest running time on ForestTypes and Reuters datasets. In addition, it is worse than BMVC on large-scale datasets Caltech256 and VGGFace2.Please explain the above phenomenon regarding running time.

**Suitability:**

2

---

### Official Review · Reviewer_VHsG · 2024-05-24

**Rating:** 5
**Confidence:** 2

**Summary:**

The paper makes contributions to the field of multi-view clustering, particularly in handling large-scale data with varying anchor quantities across views. It successfully integrates innovative clustering techniques with practical alignment strategies, enhancing both theoretical and practical aspects of multi-view clustering. However, further exploration into the scalability, dependency on initial conditions, and potential overfitting could make the approach more robust and broadly applicable.

**Strengths:**

The manuscript introduces a novel approach to multi-view clustering that addresses several challenges with existing methods. Notably, the paper proposes an automatic and aligned anchor learning strategy that optimizes the selection and alignment of anchors across different views without needing predefined numbers of clusters or anchors. This approach is particularly beneficial for handling large-scale datasets where traditional methods struggle due to computational complexity. The Hierarchical Bipartite Neighbor Clustering (HBNC) method allows for adaptive selection of anchors, potentially enhancing the clustering accuracy. Extensive experiments validate the effectiveness of the approach, indicating improvements over traditional methods in both clustering performance and computational efficiency.

**Limitations:**

Despite its strengths, the paper has several limitations. The method's performance, while superior in some respects, might still depend heavily on the initial conditions and specific characteristics of the data used in the experiments. There is also a lack of detailed discussion on the scalability of the approach when applied to even larger datasets or more complex multi-view scenarios beyond the ones tested. Additionally, the complexity of the algorithm, while reduced compared to traditional methods, might still pose challenges in terms of interpretability and ease of implementation. The paper could benefit from a clearer explanation of the mathematical formulations and a more thorough theoretical analysis to support the empirical findings. Specifically, there are the following questions:

1、The introduction effectively sets up the motivation for the study. Consider highlighting the practical implications and potential applications of the framework earlier in the introduction to engage a broader audience.

2、The description of the Hierarchical Bipartite Neighbor Clustering (HBNC) method is detailed but could be improved with a flowchart or diagram that illustrates the steps visually.

3、Expand on the criteria for dataset selection and provide more detailed statistical analysis to support the robustness of the findings. Consider using box plots or error bars in graphical results to show variability and confidence intervals.

4、Deepen the discussion on how the results compare with existing methods beyond just performance metrics. Discuss any unexpected results or anomalies in the data.

5、The conclusion could better summarize the key findings and their implications. Future work should address potential limitations and suggest specific areas for further research.

**Suitability:**

2

---

### Meta-Review · Area_Chair_mJNY · 2024-07-01

**Recommendation:** Accept (Poster)
**Confidence:** 5

**Metareview:**

The paper initially received the following ratings: BR(C1kY), WA(VHsG), BR(XUZj), BA(Ytja). After rebuttal and discussion, the final ratings were: BA(C1kY), WA(VHsG), BA(XUZj), BA(Ytja).

Below are the details of the rebuttal:
1. Reviewer C1kY and Reviewer XUZj were almost satisfied with the response and ultimately raised their ratings.

The AC carefully reviewed the rebuttal:
1. This paper is generally well-written and well-organized.
2. Regarding the contribution of the technique, the AC agrees that 3AMVC is a novel work that selects different numbers of anchors within each view by introducing and modifying hierarchical clustering. This point may provide a good perspective to inspire the design of more effective multi-view clustering algorithms in the future. Meanwhile, the AC recommends the author on their valuable insights based on the experimental results.
3. In rebuttal, the author added a supplementary experiment to a large-scale dataset and showed a good performance. This verifies the superiority of the proposed method when applied in large-scale scenarios.

In sum, after the rebuttal, the AC believes the concerns raised by the reviewers were well-addressed and recommends acceptance of this paper. In the final version, the AC strongly encourages the authors to include all discussions from the rebuttal and improve the presentation to enhance readability for a diverse audience.

---

### Meta-Review · Senior_Area_Chairs · 2024-07-10

**Recommendation:** Accept (Poster)
**Confidence:** 4

**Metareview:**

This paper received mixed ratings initially. After rebuttal, all the reviewers tend to accept the paper. SAC and AC agree with reviewers and recommend acceptance of the paper.